# Hybrid Spiking Vision Transformer for Object Detection with Event Cameras

Qi Xu [1]   Jie Deng [1]   Jiangrong Shen † [2 3 4]   Biwu Chen [5]   Huajin Tang [4 6]   Gang Pan † [4 6]

## Abstract

Event-based object detection has attracted increasing attention for its high temporal resolution, wide dynamic range, and asynchronous address-event representation. Leveraging these advantages, spiking neural networks (SNNs) have emerged as a promising approach, offering low energy consumption and rich spatiotemporal dynamics. To further enhance the performance of event-based object detection, this study proposes a novel hybrid spike vision Transformer (HsVT) model. The HsVT model integrates a spatial feature extraction module to capture local and global features, and a temporal feature extraction module to model time dependencies and long-term patterns in event sequences. This combination enables HsVT to capture spatiotemporal features, improving its capability in handling complex event-based object detection tasks. To support research in this area, we developed the Fall Detection dataset as a benchmark for event-based object detection tasks. The Fall DVS detection dataset protects facial privacy and reduces memory usage thanks to its event-based representation. Experimental results demonstrate that HsVT outperforms existing SNN methods and achieves competitive performance compared to ANN-based models, with fewer parameters and lower energy consumption.

---

[1]School of Computer Science and Technology, Dalian University of Technology, Dalian, China [2]Faculty of Electronic and Information Engineering, Xi'an Jiaotong University, Xian, China [3]National Key Lab of Human-Machine Hybrid Augmented Intelligence, Xi'an Jiaotong University, Xian, China [4]State Key Lab of Brain-Machine Intelligence, Zhejiang University, Hangzhou, China [5]Shanghai Radio Equipment Research Institute, Shanghai, China [6]College of Computer Science and Technology, Zhejiang University, Hangzhou, China. Correspondence to: Jiangrong Shen <jrshen@zju.edu.cn>, Gang Pan <gpan@zju.edu.cn>.

*Proceedings of the 42nd International Conference on Machine Learning*, Vancouver, Canada. PMLR 267, 2025. Copyright 2025 by the author(s).

## 1. Introduction

Deep neural networks have garnered considerable attention for their remarkable performance across a spectrum of tasks, ranging from computer vision to natural language processing. The exceptional performance is primarily attributed to their intricate and specialized architectures. Notably, the Transformer architecture (Beal et al., 2020; Carion et al., 2020; Guo et al., 2023) has demonstrated outstanding efficacy in object detection tasks, albeit at the expense of heightened energy consumption due to its intricate structure. In parallel, drawing inspiration from the event-driven nature of the human brain, spiking neural networks (SNNs) (Hu et al., 2023a;b; Xu et al., 2023; 2024b; Xiao et al., 2025) offer distinct advantages, including low energy consumption and rich spatiotemporal dynamics. The emergence of neuromorphic hardware such as Darwin3 (Ma et al., 2024) further supports efficient on-chip learning with a novel instruction set architecture. Recent studies have also advanced the efficiency, compressibility, and biological plausibility of SNNs (Xu et al., 2024a; Shen et al., 2025; 2024). In this context, our study explores the integration of Transformer-based Artificial neural networks (ANNs) and SNNs, aiming to combine their respective advantages in a hybrid framework for efficient object detection.

Owing to their high time resolution (in the order of microseconds), high dynamic range ($> 120dB$), and asynchronous address event representation, event cameras have shown great potential in various vision tasks (Liu et al., 2024a;b; 2025). In particular, event-based object detection has attracted increasing attention. Datasets for this task are typically collected using event cameras, such as the dynamic vision sensor (DVS) (Serrano-Gotarredona & Linares-Barranco, 2013)and ATIS sensor (Posch, 2011). The visual information of the event camera is represented in the form of an asynchronous address event stream of $\{(x, y, p, t)\}$. The event stream indicates log-luminosity contrast changes at time $t$ and position $(x, y)$. Each produced event only appears at the time and positions of a contrast change. Hence, event-based object detection has the advantage of avoiding oversampling and motion blur compared with traditional frame-based object detection. On the one hand, SNNs have shown potential for event-based object recognition with

event representation processing and less power consumption. The spike computation in SNNs quite matches the address-event representation of event data. On the other hand, ANNs with convolution and self-attention structures have shown the potential to implement event-based object detection efficiently. Hence, we focus on event-based object detection by considering the property of the event camera and combining the computation of ANNs and SNNs. Meanwhile, we collect the Fall DVS object benchmark dataset as one of the event-based object detection.

There have been some previous studies to implement event-based object detection. Graph-based vision Transformer (GET) is proposed in (Peng et al., 2023) with the event representation of graph token to utilize the temporal and polarity information of events. Graph token clusters asynchronous events based on event timestamps and polarities. Together with the Event Dual Self-Attention block and Group Token Aggregation module, GET effectively integrates spatial and temporal-polarity features. The muti-stage Recurrent Vision Transformers (RVTs) backbone is designed in (Gehrig & Scaramuzza, 2023) for object detection with event cameras. Each stage incorporates convolution components, local and sparse global attention, and recurrent feature aggregation. The convolution component downsamples the spatial resolution and acts as the conditional positional embedding for transform layers. The interleaved local and global self-attention captures both local and global features and offers linear complexity in the input resolution. The temporal recurrence is captured by long short-term memory (LSTM) cells (Hochreiter & Schmidhuber, 1997) instead of ConvLSTM cells (Liu et al., 2017), in order to minimize latency while retaining temporal information. Through the above multi-stage design, RVTs achieve fast interference and favorable parameter efficiency. Inspired by the architectures of RVTs, we explore multi-stage event-based object detection with hybrid computation based on ANNs and SNNs.

In this paper, we propose the hybrid spiking vision Transformer (HsVT) to implement event-based object detection. Firstly, we design a multi-stage vision Transformer that integrates ANN and SNN components to leverage the complementary strengths of both paradigms. Secondly, SNN components are designed to capture the temporal feature information. The self-attention and convolutional components are employed to capture the spatial features. This hybrid paradigm not only enhances temporal feature extraction through biologically spiking mechanisms but also achieves competitive performance. Thirdly, the Fall DVS detection benchmark dataset is collected in this paper as a new benchmark dataset to evaluate the performance of the proposed HsVT. Evaluated on GEN1, HsVT surpasses prior SNN-based approaches and achieves competitive performance of ANN models, while offering lower power consumption and model complexity.

## 2. Related Work

### 2.1. Event-based Object Detection Datasets

Several public datasets have been developed for event-based object detection. The Prophesees GEN1 Automotive Detection Dataset (De Tournemire et al., 2020), collected using the Prophesee GEN1 sensor (304240), provides 39 hours of driving data with annotations for pedestrians and cars. The Megapixel Automotive Detection Dataset (Perot et al., 2020), built on the high-resolution GEN4 sensor (1280720), covers diverse driving scenarios including day/night and urban/highway conditions, addressing challenges such as illumination variation and complex object classes.

The PKU-DAVIS-SOD dataset (Li et al., 2023) focuses on high-speed motion and extreme lighting conditions, offering long-term multimodal sequences for robust detection. The DSEC Detection Dataset (Gehrig & Scaramuzza, 2024) extends the original DSEC by adding labels for 60 sequences (70379 frames, 390118 boxes). These labels, generated by image-based tracking and manual correction, include both bounding boxes and class information and also track identities, for object-tracking applications.

These datasets significantly advance event-based object detection in autonomous driving scenarios. Furthermore, due to their inherent privacy-preserving nature, event cameras are well-suited for applications in healthcare and other sensitive environments.

### 2.2. Fall Detection Dataset

Fall detection is a technique that monitors and identifies whether an individual has experienced a fall, often employing visual sensors. Primarily focused on providing timely assistance, especially for the elderly or those with health vulnerabilities, the field endeavors to mitigate the serious consequences that fall can entail, including fractures, ligament tears, and other injuries. Leveraging a diverse array of sensors and technologies, such as cameras, depth cameras, and infrared sensors, fall detection systems are designed to promptly detect falls and initiate appropriate actions, such as sounding alarms, alerting paramedics or emergency services, and documenting incidents for further analysis. The application of fall detection technology extends to various domains, including healthcare, home care, and geriatric care.

In this context, the graphical representation of output bounding boxes plays a crucial role in providing users with a visual insight into the functioning of fall detection systems, proving particularly beneficial for monitoring systems, smart homes, and healthcare applications. Moreover, the utilization of bounding boxes enables the concurrent tracking of multiple targets, rendering them suitable for monitoring multiple individuals or analyzing complex environments.

Fall detection based on real-life datasets is very important for the training and performance evaluation of algorithms. However, privacy concerns can be a barrier to data sharing and disclosure (Debard et al., 2012). Among different vision sensors, the dynamic vision sensor based on spike computation could protect the facial privacy of individuals and reduce the necessary memory capacity of data storage.

## 2.3. Event-based Object Detection Methods

Early methods for event-based object detection adopt the transformed manner to convert the event data into frame-based sequences and then implement the object detection by the mature object detection models such as YOLOs (Ge, 2021). Although these methods achieve competitive performance in specific scenes, they ignore the properties and advantages of event cameras. Recently, some methods have attempted to utilize the asynchronous events directly. By employing the multi-stage backbone with spatial self-attention and convolutional modules and temporal LSTM modules, the recurrent vision Transformer model achieves quite fast interference and favorable parameter efficiency (Gehrig & Scaramuzza, 2023). Group event Transformer is proposed by decoupling the temporal-polarity information from spatial information (Peng et al., 2023). The multimodal object detection method named SODFormer is designed to leverage rich temporal cues from the event and frame visual streams (Li et al., 2023). Most of these methods use Transformer-based structures to capture the spatial and temporal features efficiently. To enhance multi-scale object detection with low energy consumption, SpikSSD (Fan et al., 2025) introduces a fully spiking backbone and a bi-directional fusion module tailored for spiking data. SpikingViT (Yu et al., 2025) introduces a transformer-based SNN detector that retains spatiotemporal features of event data via residual voltage memory and attention mechanisms, achieving strong performance in event-based object detection. In this paper, we consider the hybrid manner by combining the self-attention and convolutional modules to extract spatial features while using the energy-efficient SNN modules to extract the temporal features.

## 3. Event-based FALL Detection Benchmark Dataset

Event cameras have significant advantages over frame-based cameras in high dynamic range, low latency, and low power consumption. Event-based object detection is well-suited for applications such as autonomous driving. However, event cameras also have some disadvantages compared to frame-based cameras, such as the lack of grayscale and texture information. Therefore, we consider that the lack of grayscale information precisely protects the privacy of the subject being photographed. Many existing fall detection

datasets are either fake falls or not publicly available due to privacy concerns. We explore a new approach to fall detection datasets by using event stream data on existing fall datasets. The dataset is publicly available at: our Dropbox repository

## 3.1. Sample Processing

We convert the frame-based fall detection dataset into event-based fall detection dataset through the event camera simulator. Rebecq et al. (Rebecq et al., 2018) proposed ESIM, the first event camera simulator capable of generating large-scale and reliable synthetic event data. Unlike conventional methods that simply threshold the difference between two consecutive frames, their approach relies on tight integration with a rendering engine. Consequently, it dynamically and adaptively queries visual sample frames to accurately generate events. The basic frame-based fall dataset of Le2i fall detection dataset (Charfi et al., 2013) is employed. Each frame of the video is annotated to manually identify body positions using bounding boxes and labels the start and end frames of falls.

Hence, we utilize the event camera simulator to convert the video dataset into an event stream dataset, with the generated event data stored in 'bag' format. Each video file produces an event stream file, from which we extracted data content and converted it into an 'h5' format file.

## 3.2. Label Processing

The original dataset comprises a corresponding TXT annotation file for each video, annotating every frame of the video. The annotations include frame index, the direction of the fall, and Bounding Box (BBox) to specify the position of objects. Bbox representation follows the format $(x_1, y_1, x_2, y_2)$, where $(x_1, y_1)$ denotes the coordinates of the top-left corner of the bbox, and $(x_2, y_2)$ denotes the coordinates of the bottom-right corner of the BBox. Additionally, the frame numbers marking the onset and cessation of a fall event are annotated.

We utilize the data architectural framework of the GEN1 dataset (De Tournemire et al., 2020) and embrace a numpy schema, wherein boxes encapsulate timestamp $t$, as well as the Cartesian coordinates $x$ and $y$, denoting the location, along with the parameters $w$ and $h$, representing the width and height respectively, of the BBox. Additionally, these boxes incorporate the categorical identifier $class\_id$, the confidence measure $class\_confidence$, and the tracking identifier $track\_id$. In this context, the bounding box is delineated as $(x, y, w, h)$, wherein $(x, y)$ denotes the coordinates of the top-left vertex of the BBox, $w$ signifies the width of the BBox, and $h$ signifies the height of the box.

# 4. Methodology

## 4.1. Event Stream Representation

Event cameras operate distinctively from frame-based cameras, yielding different data formats. Unlike frame-based cameras that produce static images, event cameras generate event stream data. An event stream comprises a sequence of triplets, each representing an event, consisting of timestamps, pixel coordinates, and polarity information. The interplay between event flow and events, as well as the composition of events, is illustrated in Fig. 1 as depicted below:

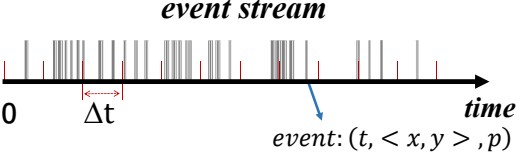

*Figure 1.* Event Stream Representation with Time Intervals. Each vertical line represents an event occurrence, while equidistant red lines represent the time intervals. Each event is represented by a triplet $(t, \langle x, y \rangle, p)$, denoting its spatial and temporal coordinates.

where $t$ represents the timestamp; $x$ and $y$ represent the pixel coordinates; $p$ represents the polarity. The timestamp signifies the time of the event, pixel coordinates denote the event's location, and polarity indicates the event's nature, with +1 for bright events and -1 for dark events. The accumulation of events that occur within a period $\Delta t$ is called a sequence E. As shown in Eq.(1), data is input into the network in the form of sequence E.

$$E = \{event_i \mid event_i \ occurs \ within \ \Delta t\}_{i=1}^{I} \quad (1)$$

In this study, we utilize three distinct datasets: GEN1, FALL Detection, and AIR Detection datasets. For the GEN1 dataset, the time interval $\Delta t_{GEN1}$ is set to 50ms, aligning with recommendations from reference (Gehrig & Scaramuzza, 2023). For the fall dataset, $\Delta t_{FALL}$ is set to be 200ms based on our visualization and experimentation detailed in Section 5.1, as illustrated in Fig 2. Finally, for the air dataset, $\Delta t_{AIR}$ is set to 10ms, taking into account the labeling frequency of 100Hz. To be specific, the configuration of $\Delta t$ in different datasets was as follows:

For the GEN1 dataset:

$$\Delta t_{GEN1} = 50ms \quad (2)$$

For the FALL dataset:

$$\Delta t_{fall} = 200ms \quad (3)$$

For the AIR dataset:

$$\Delta t_{air} = 10ms \quad (4)$$

Here, the labeling frequency is $f_{air} = 100Hz$. Hence, we set $\Delta t_{air}$ to be 10ms to ensure the sufficient feature capture of dense event data.

## 4.2. Design of HsVT

The architecture of the proposed HsVT consists of four blocks, each comprising spatial feature extraction and temporal feature extraction components. The network architecture is illustrated in Figure 2.

First, the network performs vertical propagation across stages, where each block receives input from the previous stage. Specifically, each block not only accepts input from the previous stage but also incorporates output from the temporal feature extraction module at adjacent time steps. This design enables information flow and accumulation across time, allowing the network to better capture temporal dependencies. By facilitating information sharing between neighboring time steps, the network enhances its capability to model sequential patterns in the data.

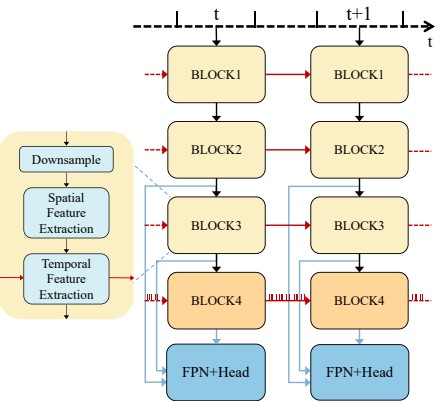

*Figure 2.* Architecture of the HsVT Network. The architecture of the proposed HsVT network consists of four blocks, each incorporating spatial feature extraction and temporal feature extraction components.

## 4.3. Spatial Feature Extraction

In each block, spatial feature extraction is composed of MaxViT (Tu et al., 2022) and SpikingMLP (Spiking Multilayer Perceptron), as illustrated in Figure 3. These components work collaboratively to extract spatial representations from the input data, enabling the network to capture both local and global spatial dependencies more effectively. The MaxViT module includes two types of self-attention mechanisms:

**Block-SA (Block-level Self-Attention).** Block-SA performs self-attention operations within local regions, capturing fine-grained spatial correlations and structural patterns. This allows the network to model spatial dependencies based

on local positional relationships, enhancing its ability to understand intra-block structures.

**Grid-SA (Grid-level Self-Attention).** Grid-SA operates across the entire feature map to model long-range dependencies and global spatial structures. Unlike Block-SA, which focuses on local features, Grid-SA integrates contextual information over larger spatial extents, enabling comprehensive global feature representation.

SpikingMLP utilises multiple layers of spiking neurons and nonlinear activation functions to further process the attention-enhanced features. By combining outputs from both Block-SA and Grid-SA, SpikingMLP extracts deeper and more abstract spatial features, thus improving the network's representational capacity.

To visualise the attention mechanisms, we present attention heatmaps generated by Block-SA and Grid-SA, respectively. Additionally, we overlay these heatmaps on the original input images for enhanced interpretability, as shown in Figure 4. The visualisations reveal that Block-SA predominantly focuses on local fine-grained regions, while Grid-SA captures broader global structures. Together, they provide complementary spatial attention patterns.

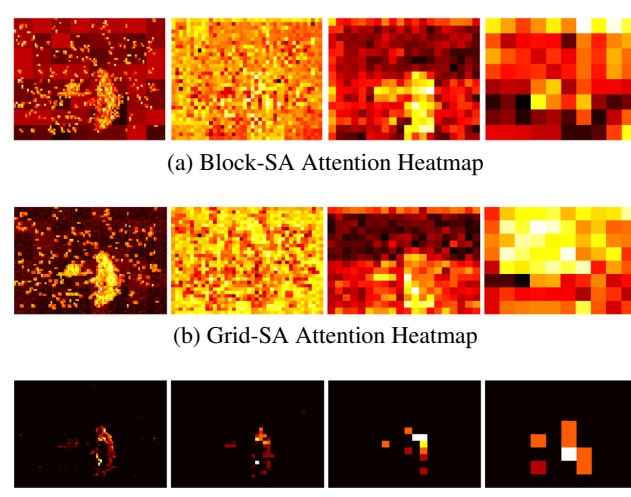

(a) Block-SA Attention Heatmap

(b) Grid-SA Attention Heatmap

(c) Overlay of Attention Maps and Input Image

*Figure 4.* Visual comparison of attention patterns from Block-SA and Grid-SA. (a) Block-SA captures local fine-grained spatial correlations. (b) Grid-SA highlights broader, long-range dependencies. (c) The overlay illustrates how these mechanisms complement each other in spatial understanding.

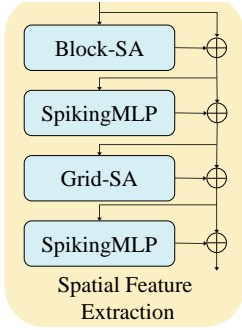

*Figure 3.* Spatial Feature Extraction in the HsVT Architecture. This figure illustrates the process of spatial feature extraction within the HsVT architecture. The components, including Block-SA, SpikingMLP, Grid-SA, and a second SpikingMLP, collaboratively work to extract spatial features from the input data.

### 4.4. Temporal Feature Extraction

In this section, we describe the utilization of LSTM and Spiking Temporal Feature Extraction (STFE) models for temporal feature extraction within the HsVT architecture, as illustrated in Figure 5.

**LSTM-Based Temporal Encoding in Initial Blocks.** In this study, LSTM serves as a crucial element for capturing both temporal dependencies and long-term dependencies within input sequences across the first three blocks. Concretely, LSTM is integrated into the architecture of each of the first three blocks. Here, it receives a fusion of spatial feature representations from the preceding block and temporal feature representations from the current block as its input. The output generated by LSTM subsequently feeds into the subsequent block, thereby enabling the propagation of temporal information throughout the entire network.

LSTM employs gating mechanisms including input gates, forget gates, and output gates to regulate the flow of information within its units. These gates control the flow of information within LSTM units, enabling them to memorize both short-term and long-term information, thus effectively handling sequential data. In the HsVT model, LSTM dynamically captures the temporal dynamics of input sequences, providing richer and more accurate temporal information during the event-based object detection process.

**Final Feature Fusion with STFE.** In the last block, we introduce the STFE model as a novel approach for temporal feature extraction. The STFE model combines the characteristics of spiking neurons and LSTM networks to capture temporal features within input sequences. Specifically, the STFE model comprises spiking neurons, convolutional layers, batch normalization layers, and a time feature extraction section similar to the traditional LSTM structure.

In the forward propagation process, the input data is extracted through the convolutional layer, and then normalized through the batch normalization layer to enhance the stability of the network. Subsequently, the normalized feature representation is fed into the spike neuron to generate a time-spike signal. Then, these spiking signals are transmitted to the STFE module for time feature extraction and propagation, in order to provide richer and more accurate



*Figure 5.* Spiking Temporal Feature Extraction in the HsVT Architecture. This figure illustrates the process of temporal feature extraction within the HsVT architecture, where the first three blocks employ the LSTM model and the final block utilizes the STFE model. This architecture integrates these models to capture the temporal dependencies in event data, facilitating the extraction of rich temporal features crucial for event-based object detection tasks.

time information for the object detection task.

In summary, we design the application of LSTM and STFE modules in temporal feature extraction and illustrate their integration and extraction of temporal information within the network. The adoption of these methods enhances the ability of the proposed model of HsVT for robust temporal feature extraction, thereby improving the performance and effectiveness of object detection tasks.

## 5. Experiments

We evaluate the proposed HsVT model through a series of experiments. First, we describe the datasets and experimental setup. Second, we perform ablation studies to validate the effectiveness of the proposed components. Third, we compare the performance of HsVT with other methods on multiple datasets. Following RVT (Table 1), we adopt the same parameters and architectural changes. Specifically, 'Channels' denotes the number of channels in each block, 'Size' represents the feature map size, 'Kernel' indicates the convolution kernel size, and 'Stride' defines the convolution stride.

*Table 1.* Parameters and Architectural Changes.

| Block | Size | Kernel | Stride | Channels | | |
| | | | | Tiny | Small | Base |
|---|---|---|---|---|---|---|
| B1 | 1/4 | 7 | 4 | 32 | 48 | 64 |
| B2 | 1/8 | 3 | 2 | 64 | 96 | 128 |
| B3 | 1/16 | 3 | 2 | 128 | 192 | 256 |
| B4 | 1/32 | 3 | 2 | 256 | 384 | 512 |

### 5.1. Datasets

The above experimental evaluations are based on three Event-Based datasets, which include those dedicated to aircraft detection, fall detection, and GEN1 Dataset.

**The Aircraft Detection Dataset.** The The Aircraft Detection Dataset consist of aircraft flighting event streams collected by DVS camera. The flighting event streams contain four different kinds of aircraft models: F1, F2, F3, and F4. This dataset is not available to the public for security reasons. Figure 6 visualizes the flighting event streams of two aircraft.

**Prophesee GEN1 Automotive Detection Dataset.** As described in section 2.1, the GEN1 dataset is collected in the autonomous driving tasks using event cameras, to detect two object categories of pedestrians and cars.

**The Fall Detection Dataset.** As described in section 2.3, The Fall Detection Dataset is generated with the event simulator by converting the original fall video dataset into event stream data. This dataset contains two different situations of fall or not fall behavior, the object detection model should predict the bounding box of humans and recognize whether fall or not.

To investigate the impact of event accumulation intervals on detection performance, we visualised reconstructed frames at three temporal resolutions: 40, 200 and 1000 ms, as shown in Figure 7. Each frame is generated by integrating events over the respective duration. As illustrated in Figure 7(a), a shorter interval of 40ms preserves finer motion details but results in sparser frames, which may limit the ability to perceive complete human contours. Conversely, Figure 7(c) with a 1000ms interval produces more cluttered representations due to excessive temporal integration, potentially leading to motion blur and ambiguity. The 200ms setting, shown in Figure 7(b), strikes a balance between temporal resolution and event density, offering clearer fall motion patterns. Quantitatively, as reported in Table 2, the model achieves the highest detection performance at 200ms, with a mean Average Precision (mAP) of 0.487.

*Table 2.* Different time intervals of Fall datasets.

| Time Interval(ms) | $mAP_{50:95}$ |
|---|---|
| 40 | 0.445 |
| 200 | 0.487 |
| 1000 | 0.405 |

### 5.2. Experimental Settings

We employ a series of optimization methods and training techniques to ensure the training effectiveness and performance of the model. Specifically, we used the RVT model combined with the YOLOX framework and maintained the

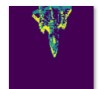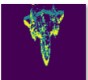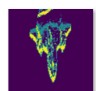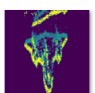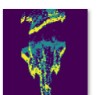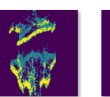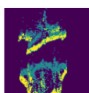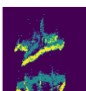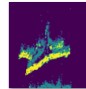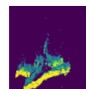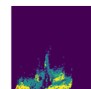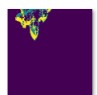

*Figure 6.* Visualization of Aircraft Detection Dataset.

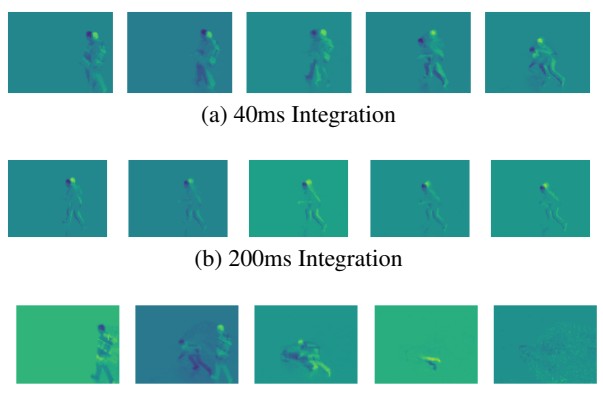

(a) 40ms Integration

(b) 200ms Integration

(c) 1000ms Integration

*Figure 7.* Event Frame Visualizations at Different Temporal Integration Intervals.

consistency of the following experimental Settings:

**Optimizers and Learning Rate Scheduling Strategies.**
We adopted the popular ADAM optimizer (Kingma & Ba, 2014) and utilized the OneCycle learning rate scheduling strategy (Smith & Topin, 2019). This strategy starts from the maximum learning rate and linearly decays during training, effectively accelerating the training speed of neural networks.

**Mixed Precision Training (Micikevicius et al., 2017).** To speed up training time and reduce memory usage without sacrificing model accuracy, we utilize mixed precision training techniques. This approach enhances training efficiency by simultaneously using low-precision and high-precision floating-point numbers, particularly suitable for handling large-scale data problems.

We train HsVT on two NVIDIA GeForce RTX 4090 GPUs, using batch size of 8 for the tiny model, and batch sizes of 4 for the basic and small model. Through these experimental settings, we train the proposed HsVT model and record the mAP value on different event-based object detection tasks.

### 5.3. Ablation Study

To validate the effectiveness of the proposed HsVT model, we conduct the ablation study to study the effect of spiking network modules with different settings in HsVT. The experiments aim to reveal the contribution of different factors of HsVT and provide important references for optimizing and improving our model. For convenience, all ablation studies

employ tiny model as the backbone of HsVT.

**Comparison of Spiking Neurons and Surrogate Functions within SpikingMLP.** As shown in Table 3, the performances of HsVT model with different spiking neuron models in the SpikingMLP component are presented. We found that the HsVT with LIF neurons performs better than IF neuron model on the AIR and FALL detection datasets. This suggests the advantage of LIF to utilize the nonlinear property to capture the spatiotemporal features in HsVT model. These findings hold practical significance for guiding the following model design and optimization processes. Hence, we employ the LIF model in the following experiments.

*Table 3.* The performance of SpikingMLP with different spike neurons.

| SpikingMLP | $mAP_{50:95}$ | |
|---|---|---|
| | FALL | AIR |
| LIFNode | 0.476 | 0.630 |
| IFNode | 0.441 | 0.587 |

We further investigate the effects of two surrogate gradient functions, ATan and Sigmoid, on the FALL and Aircraft Detection Dataset (Table 4). These functions replace the non-differentiable components of the LIF activation function, facilitating backpropagation in SNNs.

*Table 4.* The performance of Fall dataset and Air dataset with different Surrogate gradient functions.

| Surrogate function | $mAP_{50:95}$ | | |
|---|---|---|---|
| | FALL(1000ms) | FALL(200ms) | AIR |
| ATan | 0.476 | 0.490 | 0.630 |
| Sigmoid | 0.482 | 0.473 | 0.603 |

**Impact of Different SNN Components.** To explore the effectiveness of different SNN components, we evaluate several alternatives based on their parameters, computational complexity (FLOPs), and performance metrics, particularly mAP.

The considered SNN components and their corresponding characteristics are summarized in Tab. 5. Among the evaluated components, ConvBnNode(STFE) is adopted as the preferred choice due to its optimal balance between parameter efficiency and computational overhead. The combination of STFE with LIFNode neurons achieves a competitive mAP score of 0.64 with parameter counts of 0.20 and 101.19

*Table 5.* The performance of different SNN components on the Aircraft Detection Dataset.

| SNN Component | Params(M) | FLOPs(M) | Neuron | $mAP_{50:95}$ |
|---|---|---|---|---|
| STFE | 0.20 | 101.19 | IFNode | 0.595 |
| | | | LIFNode | 0.640 |
| PlainNet | 0.13 | 67.11 | IFNode | 0.594 |
| | | | LIFNode | 0.614 |
| FeedBackNet | 0.13 | 134.22 | IFNode | 0.607 |
| | | | LIFNode | 0.593 |
| StatefulSynapse | 0.13 | 67.11 | IFNode | 0.605 |
| | | | LIFNode | 0.604 |
| LSTM | 0.53 | 268.44 | - | 0.604 |

*Table 6.* Ablation study on the Aircraft Detection Dataset.

| BLOCK 1 | BLOCK 2 | BLOCK 3 | BLOCK 4 | $mAP_{50:95}$ |
|---|---|---|---|---|
| LSTM | LSTM | LSTM | LSTM | 0.595 |
| STFE | LSTM | LSTM | LSTM | 0.599 |
| LSTM | STFE | LSTM | LSTM | 0.610 |
| LSTM | LSTM | STFE | LSTM | 0.609 |
| LSTM | LSTM | LSTM | STFE | 0.640 |
| LSTM | LSTM | STFE | STFE | 0.621 |
| LSTM | STFE | STFE | STFE | 0.602 |
| STFE | STFE | STFE | STFE | 0.602 |

FLOPs, respectively. While other alternatives such as Plain-Net, FeedBackNet, and StatefulSynapseNet also present promising performance, STFE stands out for its superior parameter efficiency and computational efficacy.

These findings underscore the importance of selecting SNN components judiciously to strike the right balance between model complexity, computational cost, and performance metrics, ultimately contributing to the effectiveness and efficiency of the overall HsVT architecture.

**Impact of SNN Component Placement in Network Architecture.** Tab. 6 presents the results of ablation studies of SNN blocks with different configurations in the network structure, focusing on the influence of SNN component placement. Each row represents a different arrangement of the SNN components in the four blocks of the network structure. The mAP column shows the average accuracy of each configuration on the Air dataset. The results highlight the variation in the performance of STFE components at their location in the network structure, with mAP scoring highest when STFE components are placed entirely in the fourth block.

These results highlight the importance of placing SNN components in network architectures. Specifically, integrating STFE components into later modules tends to result in better performance, which is reflected in higher mAP scores. Interestingly, configurations with multiple STFE modules also exhibit competitive performance, indicating the potential advantages of utilizing time feature extraction at multiple stages of information processing. Taken together, these findings provide valuable insights to help optimize event-based tasks and improve SNN-based architectures.

### 5.4. Comparison with the State-of-the-art

In this section, we compare our proposed HsVT model with a wide range of state-of-the-art methods, covering different network types including CNNs, RNNs, GNNs, Transformers, SNNs, and hybrid models. Tab. 7 presents the performance comparison on the GEN1 dataset. Transformer-based methods such as ERGO-12 and STAT achieve the highest $mAP_{50:95}$ scores of 0.504 and 0.499, respectively.

Among SNN-based methods, our proposed HsVT achieves 0.478, outperforming all previous SNN, including SpikSSD (0.408), EAS-SNN (0.409), and SFOD (0.321). Although RVT (RNNs + Transformer) achieves strong performance (0.472), HsVT-B exceeds it while maintaining a smaller parameter size compared to some other large models.

As shown in Tab. 8, we further compare HsVT with RVT on the FALL dataset, as RVT represents a strong non-SNN hybrid baseline. Across all model variants (tiny, small, base), HsVT consistently outperforms RVT on both AIR and FALL categories, demonstrating superior generalization in event-based fall detection. Interestingly, we observe that increasing the model size on the FALL and Aircraft Detection datasets does not always yield performance gains, and in some cases slightly degrades accuracy. We speculate that this may be due to the limited size and diversity of these datasets, which can lead to overfitting in larger models and hinder their generalization.

## 6. Discussion and Conclusion

In this study, we propose a novel hybrid model, HsVT, which leverages the strengths of ANN and SNN for event-based object detection. We evaluated the model on the GEN1, FALL, and Aircraft Detection Dataset, demonstrating its effectiveness and efficiency.

For the GEN1 dataset, the HsVT model achieves comparable performance to the pure SNN model in terms of mAP, while requiring fewer parameters. This highlights its advantage in parameter efficiency and capability for accurate object detection. On the FALL Detection and AIR Detection datasets, the performance of the HsVT model does not strictly follow the expected Base >Small >Tiny pattern. We attribute this to dataset size limitations, which may lead to overfitting in larger models. Despite this, the HsVT model consistently outperforms the RVT model on both datasets, further validating its robustness and adaptability.

In summary, our findings underscore the effectiveness and competitiveness of the HsVT model in event-based object detection. By integrating the complementary strengths of ANN and SNN, the model achieves high performance while

*Table 7.* Performance comparison with the state-of-the-art methods on GEN1 dataset.

| Model | Network Type | Params(M) | $mAP_{50:95}$ |
|---|---|---|---|
| Inception+SSD (Iacono et al., 2018) | CNNs | - | 0.301 |
| Asynet (Messikommer et al., 2020) | GNNs | 11.4 | 0.145 |
| MatrixLSTM (Cannici et al., 2020) | RNNs+CNNs | 61.5 | 0.310 |
| RED (Perot et al., 2020) | RNNs+CNNs | 24.1 | 0.400 |
| AEGNN (Schaefer et al., 2022) | GNNs | 20.0 | 0.163 |
| ASTMNet (Li et al., 2022) | RNNs+CNNs | >100 | 0.467 |
| RVT (Gehrig & Scaramuzza, 2023) | RNNs+Transformer | 18.5 | 0.472 |
| TAF (Liu et al., 2023) | CNNs | 14.8 | 0.454 |
| ERGO-12 (Zubić et al., 2023) | Transformer | - | 0.504 |
| STAT (Guo et al., 2024) | Transformer | - | 0.499 |
| MobileNet-64+SSD (Cordone et al., 2022) | SNNs | 24.3 | 0.147 |
| VGG-11+SSD (Cordone et al., 2022) | SNNs | 12.6 | 0.174 |
| DenseNet121-24+SSD (Cordone et al., 2022) | SNNs | 8.2 | 0.189 |
| LT-SNN (Hasssan et al., 2023) | SNNs | - | 0.298 |
| EMS-ResNet34 (Su et al., 2023) | SNNs | 14.4 | 0.310 |
| SFOD (Fan et al., 2024) | SNNs | 11.9 | 0.321 |
| EAS-SNN (Wang et al., 2024) | SNNs | 25.3 | 0.409 |
| SpikeFPN (Zhang et al., 2024) | SNNs | 22.0 | 0.223 |
| KD-SNN (Bodden et al., 2024) | SNNs | 12.97 | 0.229 |
| SpikSSD (Fan et al., 2025) | SNNs | 19.0 | 0.408 |
| SpikingViT (Yu et al., 2025) | SpikingTransformer | 21.5 | 0.394 |
| HsVT-T(ours) | RNNs+SNNs+Transformer | 4.1 | 0.449 |
| HsVT-S(ours) | RNNs+SNNs+Transformer | 9.1 | 0.465 |
| HsVT-B(ours) | RNNs+SNNs+Transformer | 17.2 | 0.478 |

*Table 8.* The performance comparison on Fall dataset.

| Method | Variant | AIR | FALL |
|---|---|---|---|
| | tiny | 0.613 | 0.487 |
| RVT | small | 0.600 | 0.466 |
| | base | 0.604 | 0.469 |
| | tiny | 0.641 | 0.491 |
| HsVT | small | 0.616 | 0.492 |
| | base | 0.618 | 0.486 |

maintaining parameter efficiency. This work provides valuable information on the roles of ANN and SNN in object detection and serves as an important reference for future research in this field.

## Acknowledgements

This work was supported by the National Natural Science Foundation of China under Grants (No.62306274, 62476035, 62206037, U24B20140, 61925603), Open Research Program of the National Key Laboratory of Brain-Machine Intelligence, Zhejiang University (No.BMI2400012), the Young Elite Scientists Sponsorship Program by CAST under Grant 2024QNRC001 and Xi'an Association for Science and Technology Youth Talent Support Program No.0959202513037.

## Impact Statement

This work proposes an energy-efficient and hybrid spiking vision Transformer for event-based object detection. By leveraging the strengths of SNNs and Transformer architectures, it advances low-power spatiotemporal processing. The release of a new fall detection dataset further supports research and reproducibility in event-based vision.

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

## A. Theoretical Energy Estimation

Following the methodology described in (Zhou et al., 2022), we estimate the theoretical energy consumption of each HsVT under a 45nm process technology. For the ANN components, the energy is estimated as:

$$E_{\text{ANN}} = 4.6\,\text{pJ} \times \text{FLOPs} \qquad (5)$$

For the SNN components, the estimation incorporates spike sparsity by introducing Spike Operations (SOPs):

$$E_{\text{SNN}} = 0.9\,\text{pJ} \times \text{SOPs} = 0.9\,\text{pJ} \times fr \times T \times \text{FLOPs} \qquad (6)$$

where $fr$ denotes the average firing rate and $T$ is the total number of timesteps.

The total energy consumption of HsVT is then given by:

$$E_{\text{HsVT}} = E_{\text{ANN}} + E_{\text{SNN}} \qquad (7)$$

The experimental results of HsVT-backbone on the GEN1 dataset are shown in Table 9.

*Table 9.* Energy Consumption Analysis of the HsVT Backbone.

| HsVT-backbone | FLOPs(M) | $E_{ANN}$(mJ) | SOP(M) | $E_{SNN}$(mJ) | $E_{HsVT}$(mJ) |
|---|---|---|---|---|---|
| Tiny | 4199.59 | 19.32 | 1156.00 | 0.017 | 19.34 |
| Small | 8485.64 | 39.03 | 2413.10 | 2.172 | 41.20 |
| Base | 14229.15 | 65.45 | 3771.60 | 3.394 | 68.84 |

To analyze the energy distribution within the HsVT-based detection framework, we divide the network into three key components: the backbone, the feature pyramid network (FPN), and the detection head. We compute the theoretical energy consumption for each component individually. The results are presented in Table 10.

*Table 10.* Theoretical energy consumption (mJ) of different network components on the GEN1 dataset.

| Module | Tiny | Small | Base |
|---|---|---|---|
| Backbone | 19.34 | 41.20 | 68.84 |
| FPN+Head | 14.57 | 32.64 | 65.66 |
| Backbone +FPN+Head | 33.91 | 73.84 | 134.50 |

We present the theoretical energy consumption of our HsVT-B model and compare it with other SNN-based detectors, including SFOD and EAS-SNN-M. As summarized in Table 11, HsVT-B demonstrates a total estimated energy of 134.5 mJ, which is higher than SFOD and EAS-SNN-M. This increase is primarily attributed to the hybrid ANN-SNN structure, which integrates high-capacity ANN modules. In particular, the ANN backbone contributes 68.84 mJ, and the FPN+Head modules add 65.66 mJ. Despite the increased energy cost, the hybrid architecture achieves superior detection performance.

*Table 11.* Theoretical energy consumption comparison (mJ) on GEN1 dataset.

| Module | HsVT-B | SFOD (Fan et al., 2024) | EAS-SNN-M (Wang et al., 2024) |
|---|---|---|---|
| Architecture Type | ANN-SNN | SNN | SNN |
| Energy (Backbone) | 68.84 | – | 7.52(Embeding) + 14.12 |
| Energy (FPN + Head) | 65.66 | – | 6.46 |
| Energy (Total) | 134.5 | 7.26 | 28.10 |

