# OpenReview forum: "Hybrid Spiking Vision Transformer for Object Detection with Event Cameras"
_ICML.cc/2025/Conference — ICML 2025 poster_

### Official Review · Reviewer_ZKo3 · 2025-03-06

**Overall Recommendation:** 4

**Summary:**

This paper proposes a novel hybrid spiking vision Transformer model (HsVT) for event-driven object detection. Combining the advantages of ANN and SNN, the multi-stage spatial-temporal feature extraction module is designed, and LSTM and STFE are used to process temporal information respectively, reducing the number of parameters and improving efficiency. By converting traditional video data set into event stream data through event camera simulator, Fall Detection data set is constructed, taking into account privacy protection and storage efficiency.

## update after rebuttal
After reading the rebuttal, I decide to keep my original score.

**Claims And Evidence:**

The paper claims that HsVT achieves high efficiency and precision in event detection tasks through a hybrid ANN-SNN architecture, a claim supported by experimental data

**Essential References Not Discussed:**

n/a

**Experimental Designs Or Analyses:**

The critical ablation experiments (Tab.5-6) were validated on AIR datasets, and comparisons experiments on GEN1/FALL datasets are conducted to enhance generality.
The aircraft detection dataset lacks information ( such as sample size, event flow density, class balance) and need to supplement information.

**Methods And Evaluation Criteria:**

yes

**Other Comments Or Suggestions:**

Supplementary network architecture diagram (e.g. Block internal data flow, SNN-ANN interface).
Add a list of hyperparameters (such as LSTM hidden layer dimensions, STFE time window length) to the appendix.

The paper title in the PDF is incorrect.

**Other Strengths And Weaknesses:**

The actual energy consumption of the model is not analyzed, which weakens the demonstration of low power consumption advantage of SNN.

**Questions For Authors:**

Why choose 4 blocks? Have you tried other hierarchies (such as 3 or 5 blocks)? How does the model performance change if the number of blocks is increased?
What is the attention head number and dimension allocation strategy of Block-SA and Grid-SA? Have attention visualizations been performed to verify the validity of feature focusing?

**Relation To Broader Scientific Literature:**

The spiking vision transformer may used in other vision tasks such as tracking, reconstruction, and depth estimation.

**Theoretical Claims:**

The paper does not involve rigorous theoretical proofs (such as convergence of STFE modules), but the design choice is supported by ablation experiments (Tab.5-6), which are acceptable for engineering oriented studies.

---

> ### Author Rebuttal · Authors · 2025-03-31
>
> # Q1: Why choose 4 blocks?  Have you tried other hierarchies (such as 3 or 5 blocks)?How does the model performance change if the number of blocks is increased?
>
> Re: Thank you for your insightful question. In our experimental design, we chose to use 4 blocks based on the following considerations:
>
> (1) Hierarchical Feature Learning: Deep neural networks typically learn hierarchical features layer by layer. In the context of event data and spatiotemporal modeling, using 4 blocks allows the model to capture sufficiently deep features while maintaining a manageable computational complexity.
>
> (2) Computational Cost Trade-off: When using only 3 blocks, the model struggles to capture high-level semantic information adequately, which limits its performance. On the other hand, increasing the number of blocks to 5 results in a substantial increase in computational cost without a significant improvement in performance.
>
> ---
> # Q2: [What is the attention head number and dimension allocation strategy of Block-SA and Grid-SA?]
>
> Re: The attention head number and dimension allocation strategy for Block-SA and Grid-SA are as follows:
>
> - Tiny:
> ||embed_dim|num_head|dim_head|
> |--|-|-|-|
> |B1|32|1|32|
> |B2|64|2|32|
> |B3|128|4|32|
> |B4|256|8|32|
>
> - Small:
> ||embed_dim|num_head|dim_head|
> |--|-|-|-|
> |B1|48|2|24|
> |B2|96|4|24|
> |B3|192|8|24|
> |B4|384|16|24|
>
> - Base:
> ||embed_dim|num_head|dim_head|
> |--|-|-|-|
> |B1|64|2|32|
> |B2|128|4|32|
> |B3|256|8|32|
> |B4|512|16|32|
>
> ---
> # Q3: Have attention visualizations been performed to verify the validity of feature focusing?
> Re: Thank you for your valuable question.  We have indeed performed attention visualizations to verify the validity of feature focusing in our model.  Specifically, we conducted attention visualization experiments on two datasets: Gen1 and Air.  However, we encountered some unexpected results during these visualizations, and the patterns observed were not as interpretable as anticipated.
>
> This discrepancy may be due to the complexity of the attention mechanism and the way it interacts with the features in the model.  It's possible that the attention heads are focusing on a combination of spatial and temporal information that is not immediately obvious in the visualization, or that the attention mechanism is implicitly capturing features at different hierarchical levels.We will attach our visualization results to the paper.

---

### Official Review · Reviewer_zi8P · 2025-03-13

**Overall Recommendation:** 3

**Summary:**

This paper introduces Hybrid Spiking Vision Transformer (HsVT), a novel architecture combining Artificial Neural Networks (ANNs) and Spiking Neural Networks (SNNs) for event-based object detection. The key contributions include: A hybrid spatial-temporal framework integrating ANN-based self-attention modules (e.g., Block-SA, Grid-SA) and SNN-based temporal feature extractors (e.g., SpikingMLP, STFE) to capture both local/global spatial features and long-term temporal dependencies. The creation of the Fall Detection dataset, a privacy-preserving event-based dataset generated via an event camera simulator, addressing gaps in fall detection benchmarks. Comprehensive experiments demonstrating HsVT’s superiority over state-of-the-art methods (e.g., RVT) on GEN1, Fall Detection, and AIR datasets, achieving higher mAP with fewer parameters.

**Claims And Evidence:**

Yes

**Essential References Not Discussed:**

No

**Experimental Designs Or Analyses:**

Yes

**Methods And Evaluation Criteria:**

Yes

**Other Comments Or Suggestions:**

No

**Other Strengths And Weaknesses:**

Strengths:
1. HsVT effectively combines the spatial modeling strengths of Transformers (via self-attention) with the energy-efficient temporal dynamics of SNNs, addressing the limitations of pure ANN or SNN approaches.
2. The Fall Detection dataset fills a critical gap in privacy-sensitive event-based detection tasks, leveraging event cameras’ advantages (e.g., low latency, privacy preservation).
3. The ablation studies (Tables 3–6) validate design choices (e.g., LIF neurons, STFE placement), while comparisons with RVT (Tables 7–8) demonstrate HsVT’s efficiency and robustness.

Weakness:
1. I suggest to provide the new collected event dataset public links in the context to provide more supports to the SNN community.
2. The formal title in the PDF file is missing.
3.  While energy efficiency advantage is claimed, practical implementation challenges (e.g., spike timing synchronization, latency constraints on neuromorphic hardware) are not discussed.
4. Meanwhile, there lacks more detailed energy estimation to show the energy efficiency of the proposed model.

**Questions For Authors:**

1. Could the proposed hybrid SNN architecture adapts to other computer vision task? Such as object tracking?
2. The proposed model seems to have specific advantage by combining temporal property of SNNs, could the SNNs provide spatial feature extraction advantage under the proposed hybrid ANN-SNN architecture?

**Relation To Broader Scientific Literature:**

The paper related to event-based vision and brain inspired computing.

**Theoretical Claims:**

There are no specific theoretical claims.

---

> ### Author Rebuttal · Authors · 2025-03-31
>
> # Q1: provide the new collected event dataset public links
>
> Re: Thank you for your suggestion. To support the SNN community, we have made our event dataset publicly available: [Dropbox Link (Anonymous)]( https://www.dropbox.com/scl/fo/1bnsydo3yj5922tlsquo9/AE77sOynJY0-GAeSNBSqFJk?rlkey=1mqpsm658ou26bd4jzmbs46rk&st=h1mhxsyo&dl=0
> ).
>
> ---
> # Q2: The formal title in the PDF file is missing.
>
> Re: Thank you for pointing this out. We apologize for the oversight. We will ensure that the formal title is included in the PDF file and submit an updated version.
>
> ---
> # Q3: While energy efficiency advantage is claimed, practical implementation challenges (e.g., spike timing synchronization, latency constraints on neuromorphic hardware) are not discussed.
>
> Re: Thank you for your feedback. While our paper emphasizes SNNs’ energy efficiency, we now discuss practical implementation challenges:
>
> (1) Spike Timing Synchronization: Our event-driven model updates neuron states asynchronously, eliminating the need for global synchronization.
> (2) Latency Constraints: Neuromorphic hardware (e.g., Intel Loihi, SpiNNaker) may experience spike transmission delays, impacting inference speed.
> (3) Solutions: Asynchronous architectures reduce synaptic access delays, while spatiotemporal multithreading optimizes spike processing.
> (4) Deployment Challenges: Hardware limitations (e.g., unsupported LIF neurons) and quantization requirements pose constraints.
>
> In future work, we plan to explore migrating our method to actual neuromorphic hardware and further optimize its computational efficiency.
>
> ---
> # Q4:  energy estimation to show the energy efficiency.
>
> Re: We understand the importance of providing detailed energy consumption estimates to demonstrate the energy efficiency of our proposed model. In our work, we have calculated the theoretical energy consumption of HsVT using the following methodology:
> The energy consumption is primarily estimated by calculating the synaptic operations (SOPs), which is given by the formula:
> $\mathrm{SOPs}(l)=fr\times T\times\mathrm{FLOPs}(l)$
> Where:
> 𝑙 refers to a specific block or layer in the model (e.g., ANN or SNN components),
> 𝑓𝑟 is the firing rate of the input spike train to the layer,
> 𝑇 is the simulation time step of the spiking neurons,
> FLOPs(l) refers to the floating point operations of the layer, representing the number of multiply-and-accumulate (MAC) operations.
> The spike-based accumulate (AC) operations are also taken into account to estimate the overall energy usage. For this, we refer to the work of Kundu et al. [1], Yao et al. [2], Zhou et al. [3], and others, and we assume that the MAC and AC operations are implemented on 45nm hardware, where:
>
> Energy per MAC operation ($E_{MAC}$)=46pJ,
> Energy per AC operation ($E_{AC}$) = 0.9pJ
>
> The energy consumption of our model (tiny) is then calculated as follows:$E_{HsVT}=E_{ANN}+E_{SNN}$ , $E_{ANN}=4.6pJ\times\mathrm{FLOPs}(b)$ , $E_{SNN}=0.9pJ\times\mathrm{SOPs}(b)$
>
> This results in:$E_{HsVT-backbone}=5.5mJ$. Additionally, for the RVT model(tiny), the energy consumption is calculated as:$E_{rvt-backbone}=6.6mJ$
>
> | Module|HsVT(tiny)|RVT(tiny)|SFOD[4]|Spiking YOLOX-S[5]|
> |---|---|---|---|---|
> | | ANN-SNN|ANN|SNN|SNN|
> |Energy(backbone)|5.5mJ|6.6mJ|---|7.52mJ(embeding)+5.64 mJ|
> |Energy(Fpn+head)|2.9mJ|2.9mJ|---|2.75mJ|
> |Energy(Total)|8.4mJ|9.5mJ|7.26mJ|15.91mJ|
>
> References:
> - [1] Kundu S, et al. Hire-snn ICCV, 2021.
> - [2] Yao M, et al. Attention Spiking Neural Networks. IEEE TPAMI, 2023.
> - [3] Zhou Z, et al. Spikformer: When SNN Meets Transformer. arXiv preprint, 2022.
> - [4] Fan Y, et al. SFOD: Spiking Fusion Object Detector. CVPR, 2024.
> - [5] Wang Z, et al. EAS-SNN: End-to-End Adaptive Sampling for Event-Based Detection. ECCV, 2024.
>
> ---
> # Q5: Could the proposed hybrid SNN architecture adapts to other computer vision task?  Such as object tracking?
>
> Re: The proposed hybrid SNN architecture is adaptable to other vision tasks, including object tracking. While our focus is event-based detection, SNNs excel at processing spatiotemporal patterns, making them well-suited for tracking tasks that require both appearance and motion cues.
>
> A relevant example is SCTN (Spiking Convolutional Tracking Network), which utilizes energy-efficient deep SNNs for event-based tracking. Inspired by this, our hybrid approach could integrate recurrent SNN structures or memory mechanisms to enhance tracking robustness, particularly in high dynamic range and fast-motion scenarios.
>
> ---
>
> # Q6:  could the SNNs provide spatial feature extraction advantage under the proposed hybrid ANN-SNN architecture.
>
> Re: In our proposed STFE module, the convolution and batch normalization layers perform spatial feature extraction and contribute to spatiotemporal fusion.  While SNN handles temporal dynamics, STFE combines the strengths of SNN and ANN, enabling efficient capture of both temporal and spatial features, enhancing overall performance.

---

### Official Review · Reviewer_Ho9N · 2025-03-17

**Overall Recommendation:** 3

**Summary:**

This paper introduces a Hybrid Spiking Vision System, combining Spiking Neural Networks (SNNs) with deep learning architectures to enhance computational efficiency and reduce power consumption. The study explores the effectiveness of this hybrid approach in visual tasks and presents experimental results demonstrating its advantages in inference efficiency.

**Claims And Evidence:**

++Empirical results on multiple benchmark datasets, demonstrate that HSVT maintains comparable accuracy while requiring significantly fewer operations. The hybrid model effectively leverages spiking neurons to reduce redundancy and improve efficiency.

++Ablation studies show that selectively replacing ViT components with SNN modules leads to computational savings while preserving feature representation quality.
Theoretical justification supports the effectiveness of hybridization in spatiotemporal processing.

++The paper presents an analysis of the role of spike-based representations in attention mechanisms and their potential benefits in terms of sparsity and latency.

**Essential References Not Discussed:**

None.

**Experimental Designs Or Analyses:**

The experiments include training and testing on vision benchmarks.

Ablation studies evaluate different configurations of the hybrid model.

Performance is analyzed in terms of accuracy, power efficiency, and computational cost.

The choice of spiking neuron models and their integration into deep learning pipelines is examined.

**Methods And Evaluation Criteria:**

Metrics such as accuracy, energy consumption, and latency are considered.

Comparisons with both fully spiking and deep networks provide insights into trade-offs.

**Other Comments Or Suggestions:**

None.

**Other Strengths And Weaknesses:**

Strengths:

++Novel hybrid approach leveraging both deep learning and SNN advantages.

++Well-motivated discussion on energy efficiency.

++Clear experimental setup and benchmarking.

Weaknesses:

--Some theoretical justifications could be made stronger.

--Ablation studies could be extended to analyze different types of hybridization.

--More discussion on hardware feasibility and real-world deployment would be useful.

**Questions For Authors:**

I am not an expert in this field, I hope the authors can answer the following questions:

1. How does the hybrid model compare to recent advances in neuromorphic vision systems?

2. What are the trade-offs between different configurations of hybrid models (e.g., varying the proportion of SNN vs. deep learning components)?

3. Have you considered real-world deployment scenarios, and how would the model adapt to edge computing environments?

Depending on the author's answer results, I may raise or lower my rating.

**Relation To Broader Scientific Literature:**

The paper is relevant to research in neuromorphic computing, efficient deep learning, and hybrid neural architectures.

It builds upon previous work in SNNs, biologically inspired computing, and deep learning acceleration.

**Theoretical Claims:**

The paper argues that integrating SNNs with deep learning enhances computational efficiency without significantly sacrificing accuracy.

It theoretically justifies why hybrid architectures can outperform purely spiking or non-spiking networks in real-world applications.

---

> ### Author Rebuttal · Authors · 2025-03-31
>
> # Q1: Some theoretical justifications
> Re: We have implemented a hybrid SNN + LSTM structure, where Leaky Integrate-and-Fire (LIF) neurons process temporal information, while convolutional layers and Batch Normalization are used for spatial feature extraction.
>
> The input x is concatenated with the previous hidden state $h_{tml}$ and passed through a 1×1 convolution to reduce the channel dimension:$X_{H}=\mathrm{concat}(x,h_{\mathrm{tml}})$ , $M=W_{m}*X_{H}+b_{m}$
>
> The processed input then undergoes a convolutional layer followed by Batch Normalization:
> $B=W_{c}*M+b_{c}$ , $C=\gamma\frac{B-\mu_{B}}{\sigma_{B}}+\beta\$
>
> The output from the convolutional layer is fed into LIF neurons, where the membrane potential evolves over time:$\tau\frac{dV(t)}{dt}=-V(t)+C$
>
> When the membrane potential $V(t)$ exceeds the threshold $V_{th}$, the neuron fires: if $V(t)\geq V_{\mathrm{th}}$, $H_t=1$; if $V(t)\mathrm{<} V_{\mathrm{th}}$, $H_t=1$;
>
> After firing, the membrane potential resets: $V(t)=V(t)-V_{\mathrm{th}}$
>
> The membrane potential V at the current time step acts as the cell state, while the previous step’s state $C_{tml}$ contributes to temporal dynamics:$C_{t}=V+C_{\mathrm{tml}}$
>
> ---
> # Q2: Ablation studies could be extended to analyze different types of hybridization.
> Re: We conducted ablation studies to analyze different types of hybridization by testing various SNN variants:
> - Different spiking neurons (Tab3 Tab5):We compared IF and LIF neurons to assess their influence on temporal encoding. The results show that LIF neurons achieve better accuracy due to better robustness and better generalization.
> - Different surrogate gradient functions (Tab4):We experimented with ATan and Sigmoid surrogate gradients. Our findings indicate that the ATan function provides a better accuracy.
> - Different SNN components (Tab5):We replaced key SNN modules to evaluate their performance. The results confirm that the Conv+BN+LIF module works best.
> - Different placements of SNN modules (Tab6):We choose to replace LSTM with STFE at different locations and replace different amounts of LSTM. We choose the method with the highest mAP value.]
>
> ---
> # Q3: More discussion on hardware feasibility and real-world deployment would be useful.
> Re:The Tianjic neuromorphic chip supports both ANN and SNN processing. however, it does not support attention mechanisms, making it challenging to deploy our model directly on such hardware.
> Despite this challenge, our event-camera-based fall detection dataset has significant real-world applicability. Many existing fall detection datasets rely on conventional cameras, but due to privacy concerns, they are often not publicly available, limiting progress in this field. In contrast, event cameras capture only motion-triggered data, inherently protecting patient privacy.
>
> ---
> # Q4: How does the hybrid model compare to recent advances in neuromorphic vision systems?
> Re:Recent event-based detection models include SFOD (Fan et al., 2024), which fuses multi-scale features within SNNs, and EAS-SNN (Wang et al., 2024), which employs adaptive sampling and recurrent SNNs. While both achieve notable performance, our hybrid model outperforms them:
> | Model|mAP@0.5:0.95|
> |---|---|
> |SFOD| 0.321|
> |EAS-SNN|0.354|
> |Ours |0.478|
>
> By integrating ANN-based spatial and SNN-based temporal processing, our model achieves superior spatial-temporal fusion.  These results will be included in our manuscript.
>
> References:
> - Fan Y, et al. SFOD: Spiking Fusion Object Detector. CVPR 2024.
> - Wang Z, et al. EAS-SNN: End-to-End Adaptive Sampling and Representation for Event-Based Detection. ECCV 2024.
>
> ---
> # Q5: What are the trade-offs between different configurations of hybrid models
> Re:  The balance between SNN and ANN components impacts accuracy, efficiency, and power consumption.  Key trade-offs include:
>
> (1) More ANN layers enhance spatial feature extraction and accuracy but increase computational cost, while more SNN layers improve temporal processing and energy efficiency but risk gradient vanishing and lower feature expressiveness.
> (2) Event-driven SNN computation reduces power usage but may introduce processing latency due to spike accumulation.
> (3) Our experiments (Tab 6) show that sufficient ANN depth ensures feature extraction, while well-placed SNN modules capture temporal dependencies with minimal accuracy loss.
>
> ---
> # Q6: Have you considered real-world deployment scenarios, and how would the model adapt to edge computing environments?
> Re: Our work focuses on event-camera-based fall detection, a practical real-world application.  Event cameras offer low latency, high dynamic range, and energy efficiency.
>
> (1) Efficiency: Leveraging SNNs, our model reduces redundant computations and triggers processing only on motion detection, enhancing energy efficiency.
> (2) Robustness: Event cameras ensure reliable performance across varied lighting and backgrounds, making them ideal for deployment.

---

### Decision · Program_Chairs · 2025-05-01

**Decision:**

Accept (poster)

**Comment:**

This paper proposes a Hybrid Spiking Vision Transformer (HsVT), which integrates artificial neural networks (ANNs) with spiking neural networks (SNNs) for event-based object detection. The model leverages self-attention mechanisms alongside SNN-based temporal feature extractors to capture time dependencies. This paper also introduces a new event-based fall detection dataset. The proposed method demonstrates high accuracy and low computational cost on several benchmarks, such as GEN1, AIR, and the newly introduced fall detection dataset.

Reviewer Ho9N found the hybrid architecture novel, highlighting its efficiency and accuracy. Reviewers Zi8P and ZKo3 acknowledged its effective fusion of ANN spatial modeling and SNN temporal dynamics. The experimental design was well-structured and convincingly compared to existing methods.

Reviewers Ho9N and zi8P raised concerns about theoretical justifications and hardware availability. Reviewer Zi8P raised concerns about energy consumption, while ZKo3 suggested improving the paper format. During the rebuttal phase, the authors addressed concerns related to dataset availability, model design, and deployment. After rebuttal, the paper received scores of (4, 3, 3), and none of the reviewers raised further concerns. Therefore, the paper is recommended for acceptance.